# Comparative Analysis of the Effect of Gamma-, Electron, and Proton Irradiation on Transcriptomic Profile of *Hordeum vulgare* L. Seedlings: In Search for Molecular Contributors to Abiotic Stress Resilience

**DOI:** 10.3390/plants13030342

**Published:** 2024-01-23

**Authors:** Alexander Prazyan, Mikhail Podlutskii, Polina Volkova, Elizaveta Kazakova, Sofia Bitarishvili, Ekaterina Shesterikova, Vyacheslav Saburov, Ekaterina Makarenko, Maria Lychenkova, Marina Korol, Evgeniy Kazakov, Alexander Moiseev, Stanislav Geras’kin, Ekaterina Bondarenko

**Affiliations:** 1Russian Institute of Radiology and Agroecology of National Research Centre “Kurchatov Institute”, 249035 Obninsk, Russia; 2Independent Researcher, 2440 Geel, Belgium; 3A. Tsyb Medical Radiological Research Centre—Branch of the National Medical Research Radiological Centre of the Ministry of Health of the Russian Federation, 249036 Obninsk, Russia

**Keywords:** plant radiobiology, common barley, abiotic stress resilience, seedling irradiation, ionising radiation, gamma-rays, electrons, protons, reactive oxygen species

## Abstract

The development of adaptation strategies for crops under ever-changing climate conditions is a critically important food security issue. Studies of barley responses to ionising radiation showed that this evolutionarily ancient stress factor can be successfully used to identify molecular pathways involved in adaptation to a range of abiotic stressors. In order to identify potential molecular contributors to abiotic stress resilience, we examined the transcriptomic profiles of barley seedlings after exposure to γ-rays, electrons, and protons. A total of 553 unique differentially expressed genes with increased expression and 124 with decreased expression were detected. Among all types of radiation, the highest number of differentially expressed genes was observed in electron-irradiated samples (428 upregulated and 56 downregulated genes). Significant upregulation after exposure to the three types of radiation was shown by a set of ROS-responsive genes, genes involved in DNA repair, cell wall metabolism, auxin biosynthesis and signalling, as well as photosynthesis-related genes. Most of these genes are known to be involved in plant ROS-mediated responses to other abiotic stressors, especially with genotoxic components, such as heavy metals and drought. Ultimately, the modulation of molecular pathways of plant responses to ionising radiation may be a prospective tool for stress tolerance programmes.

## 1. Introduction

In the modern world, where food security is paramount, increasing crop yield has become a critically important task [1,2,3]. Changing climatic conditions will impose additional stress exposure on agricultural plants, such as increasing drought, salinity, and phytopathogens metabolic activity. Therefore, developing adaptation strategies for crops under those challenging growth conditions is of utmost importance. One of the most complex tasks in crop improvement is elucidating the signal transduction pathways and how they are activated and respond to different stressors [4]. As an evolutionarily ancient stress factor [5], ionising radiation (IR) is a promising tool for studying plant adaptation mechanisms to abiotic stress factors [6,7,8]. An important aspect of plant responses to IR is the activation of antioxidant defence and DNA repair mechanisms [9,10,11,12]. High doses or long-term (chronic) irradiation can cause physiological damage, resulting in decreased photosynthetic activity, suppressed metabolic processes, and, ultimately, disruption of normal plant development [9,13,14,15,16,17], which is often attributed to the genotoxic effects of γ-radiation [8,10,11,18].

One of the most extensively studied agricultural species is common barley (*Hordeum vulgare* L.), an important cereal crop used in various food industry sectors, especially for bread and beer production [19]. Despite high nutritional value and adaptability, the maximal productivity of this crop is limited by climate changes, anthropogenic pollutants, and the presence of pathogens and phytotoxins [20,21,22]. Understanding the mechanisms of barley adaptation to extreme conditions can contribute to developing new methods for enhancing yield and resilience in this cereal.

Radiobiological studies of barley responses to IR showed that this stress factor can be successfully used to identify molecular pathways involved in barley adaptation to abiotic stressors [23,24,25]. Numerous papers are focused on barley’s adaptive responses to γ-radiation, as it is the radiation type more available for research [9,23,24,25,26,27,28], where growth-stimulating and growth-inhibiting dose-dependent effects are considered in molecular details. Several studies of proton and electron radiation effects on barley plants are also available in the literature, demonstrating both positive and negative impacts on its growth, development, and productivity, alone or in combination with other stress factors [29,30]. The effects are usually attributed to the direct (DNA damage) or indirect (reactive oxygen species (ROS) production) action of ionising radiation.

Common environmental pollutants and other abiotic stress factors also possess genotoxic properties. For instance, polycyclic aromatic hydrocarbons, relatively chemically inert compounds, through metabolic activation of electrophilic derivatives, are capable of covalently interacting with nucleophilic centres of DNA and induce base pair substitutions, frame-shift mutations, deletions, S-phase arrest, DNA strand breaks, and various chromosomal alterations [31,32,33]. Drought can provoke stress that activates genes involved in antioxidant functions and DNA repair [34], including those involved in nucleotide and base excision repair pathways [35]. Heavy metals, such as chromium, lead, cadmium, and arsenic, also exhibit genotoxic effects, triggering apoptosis in plant cells and increasing the frequency of aberrant cells [36,37,38,39]. Having in mind that adaptation pathways to many stress agents have similar mechanisms [39,40], it is plausible to use ionising radiation as a tool for revealing potential molecular contributors to abiotic and anthropogenic stress resilience.

In the current work, we used three different IR types with low linear energy transfer (LET, gamma, electrons, and protons) and whole-transcriptome sequencing to reveal commonly responsive molecules and to identify if those molecules are involved in responses to other abiotic stress factors. Ionising radiation is a convenient tool for such research as its dose, intensity, and homogeneity of tissue exposure are easier to control than other stress types.

## 2. Results

### 2.1. Overview of Differential Gene Expression for the Three IR Types

A total of 553 unique differentially expressed genes (DEGs) with increased expression and 124 DEGs with decreased expression were detected compared to the control (Figure 1 and Figure 2, Appendix A).

In samples subjected to γ-irradiation, 119 DEGs were found, with 104 genes upregulated and 15 downregulated (Figure 1A, Appendix A). A significant change in the transcriptional profile was also observed in proton-irradiated samples. A total of 409 DEGs were detected, with 338 genes showing upregulation and 71 genes presenting downregulation compared to the control group (Figure 1B, Appendix A). Finally, 484 differentially expressed genes were observed in samples subjected to electron irradiation, the highest number among all types of radiation. Among them, 428 were upregulated, and 56 were downregulated (Figure 1C, Appendix A).

### 2.2. Overlapping Transcriptional Responses among Different Types of IR

The pairwise comparisons revealed DEGs unique for each radiation type, as well as transcripts involved in the overlapping response to all three types of IR. All identified genes are presented in Figure 2, Appendix A.

### 2.3. γ-Radiation/Electrons and γ-Radiation/Protons Pairwise Comparisons

Fifty-nine DEGs were identified as overlapping between γ-radiation and electron exposures (Figure 2A). Among them, two genes, *HORVU.MOREX.r3.1HG0073500* and *HORVU.MOREX.r3.2HG0206630*, had the highest values of differential expression in both conditions (|log_2_FC| > 2) (Appendix A). *HORVU.MOREX.r3.1HG0073500* encodes laccase, which is involved in lignin decomposition and the detoxification of lignin-derived products, while *HORVU.MOREX.r3.2HG0206630* encodes a protein containing the Rx_N domain, often found in plant resistance proteins [41]. Among other interesting upregulated transcripts is a gene, *HORVU.MOREX.r3.5HG0512410*, homologous to a negative regulator of p53 (Appendix A). The equivalent of p53 in plants is SUPPRESSOR OF GAMMA RESPONSE 1 (SOG1) [42], and upregulation of its negative regulator may reflect control of double-strand break plant responses. Significant downregulation was observed for three transcripts (Figure 2B, Appendix A): *BaRT2v18chr7HG356510*, *HORVU.MOREX.r3.6HG0545390* (*BaRT2v18chr6HG285660*), and *HORVU.MOREX.r3.6HG0539130*, which encode membrane protein kinases.

The comparison of transcriptional responses of γ- and proton irradiation revealed 61 upregulated DEGs (Figure 2C, Appendix A). The highest differential expression values (|log_2_FC| > 2) were observed for the same two genes, *HORVU.MOREX.r3.1HG0073500* and *HORVU.MOREX.r3.2HG0206630*, as for gamma/electron comparison. However, three downregulated genes were different (Figure 2D, Appendix A): *HORVU.MOREX.r3.4HG0418190* encodes a metacaspase that plays a key role in apoptosis; *HORVU.MOREX.r3.1HG0094920* encodes flavonoid 3′-monooxygenase, catalysing the oxidation of flavonoids in cells; *HORVU.MOREX.r3.3HG0285150* is presumed to encode a protein associated with tryptophan aminotransferase.

The GO enrichment analysis results for pairwise comparisons of γ- and electrons and γ- and protons exposures were similar for upregulated DEGs (Appendix A; Appendix A). For example, when considering the molecular function (MF) group (Appendix A), protein folding chaperone, single-stranded RNA binding, phosphate group as acceptor, pseudouridine synthase, beta-galactosidase, and phosphotransferase activities were almost equally enriched under the mentioned experimental conditions. However, MF in barley irradiated with γ-rays and electrons showed a high degree of enrichment in genes involved in phenylalanine ammonia-lyase activity, pyruvate kinase activity, hydroquinone:oxygen oxidoreductase activity, while barley irradiated with γ-rays and protons had DEGs associated with transferase activity, transferring aldehyde or ketonic groups. Similar levels of enrichment were observed in terms of cellular component (CC) in the γ- and protons and γ- and electrons comparisons for chromatin, apoplast, cell wall, and ribosome and plastid stroma (Appendix A). Enrichment of genes related to microtubules was also shared by γ- and protons-irradiation groups (Appendix A). Conversely, in the γ- and electrons condition, a high level of enrichment was found in the plastid-encoded RNA polymerase complex (Appendix A). Regarding biological processes (BP) (Appendix A), the highest enrichment levels were found for the GTP metabolic process. In plants irradiated with protons and γ-rays, a high level of enrichment was also presented in the regulation of the photosynthesis light reaction (Appendix A), while in samples irradiated with γ-rays and electrons, the cinnamic acid biosynthetic process group was highlighted (Appendix A). Among downregulated genes under γ- and proton irradiation conditions, enrichment was identified for a group of carbon-sulphur lyase activity; in the plants exposed to γ-rays and electrons, enrichment was detected for polysaccharide binding.

### 2.4. Proton and Electron Pairwise Comparison

The highest number of upregulated genes was shared by proton and electron treatments. Two hundred forty-four genes were upregulated (Figure 2E, Appendix A), and 12 genes (Figure 2F, Appendix A) were downregulated. Among downregulated genes, the lowest expression was recorded for *HORVU.MOREX.r3.4HG0346410* encoding monogalactosyldiacylglycerol synthase, which is involved in the normal functioning of the photosynthetic system, maintenance of chloroplast membrane integrity, and protection of plants from stress conditions, and *HORVU.MOREX.r3.3HG0296450* encoding glycerophosphodiester phosphodiesterase, which participates in the regulation of signalling pathways and plays a role in photosynthesis and phosphorus metabolism.

The total number of commonly expressed genes under proton and electron irradiation was 256 (Figure 2E,F), which is several times higher compared to γ-radiation-induced changes (Appendix A), and a large number of highly specific GO terms were enriched in terms of MF dictionary, such as ferrous iron binding, glutathione binding, xylan-1,4-beta-xylosidase, DNA topoisomerase type II (double strand cut, ATP-hydrolysing), pyruvate dehydrogenase a, tripeptide transmembrane transporter, acid-amino acid ligase, and acyl carrier activities (Appendix A). At the CC dictionary, a glycerol-3-phosphate dehydrogenase complex was noted (Appendix A). In terms of BP, regulation of oxidoreductase activity, regulation of cell shape, threonine biosynthetic process, NADH oxidation, cotyledon development, dipeptide transmembrane transport, and photosystem I assembly were of particular interest, largely due to the fact that these groups were specific and were not observed under γ-radiation treatment (Appendix A).

Downregulated DEGs, common for electron and proton treatments, revealed several enriched GO terms, including UDP-galactosyltransferase, glycerophosphodiester phosphodiesterase, protein tyrosine kinase activities (MF), the salicylic acid biosynthetic process, anion homeostasis, cellular response to cold, and the alditol metabolic process (BP) (Appendix A).

### 2.5. Overlapping Response to Three Types of Ionising Radiation Exposure

When comparing transcriptional responses to the three types of irradiations, 47 common DEGs were identified (Figure 2G,H), all upregulated, and 15 encoded various ribosomal proteins and translation-related transcription factors (Table 1).

To identify common response patterns to different types of radiation, the GO enrichment analysis was performed based on the DEGs in each treatment type (Appendix A). According to the annotation, at the CC level, 10 enrichment groups can be distinguished (Appendix A). The highest enrichment was observed in the nuclear chromatin group and genes associated with apoplast, followed by a group related to the cell periphery, within which more specific genes related to the plant cell wall can be identified. When considering the molecular function (Appendix A), genes involved in chaperone folding, beta-galactosidase activity, phosphotransferase activity, and phosphate group as acceptor were noted. At the same time, the highest enrichment was observed in the cellulose synthase activity and single-stranded RNA binding groups.

## 3. Discussion

Ionising radiation, including γ-radiation, electrons, and protons, converts neutral atoms or molecules into reactive ions, triggering various biological effects and targeting plant signalling systems [43,44]. The IR types studied here can be considered as low linear energy transfer radiation, with relative biological effectiveness close to 1. However, the penetration ability of these IRs is different, and their effects on plants vary on morpho-anatomical, biochemical, and molecular levels [5,12,45,46]. Depending on the dose applied, IR might be used as a tool for abiotic stress mitigation, potentially enhancing nutrient uptake, secondary metabolite biosynthesis, and osmolytes [47]. Therefore, the modulation of molecular pathways of IR responses may be a prospective tool for plant abiotic stress tolerance programmes.

In order to identify molecular pathways involved in barley adaptation to abiotic stressors and reveal potential molecular contributors to abiotic stress resilience, we classified DEGs common for the three types of IR into groups of most typical responses to ionising radiation exposure. All genes were distributed into 20 groups based on the functions of their protein products (Appendix A). Next, we discuss those DEGs known to respond to abiotic stress factors other than IR.

One of the major consequences of exposure to any ionising radiation is the increase in ROS production [12,26], leading to oxidative stress and the increased activity of the antioxidant system [23]. Oxidative stress may also be induced by other genotoxic agents, such as excess copper [48], cadmium [49,50,51], lead [52], zinc [53], as well as polycyclic aromatic hydrocarbon pollution [54], and benzene stress conditions [55]. The accumulation of ROS that are detrimental to plant growth and development is induced by such abiotic stresses as drought and salinity [56]. Excessive production of ROS, including hydrogen peroxide, poses a threat to plant growth and reproduction [57]. In plants, hydrogen peroxide affects the expression of a large set of genes involved in various growth aspects and response to environmental stimuli [58]. To regulate hydrogen peroxide balance, plants synthesise various enzymes with peroxidase activity, such as ascorbate peroxidase [59] and glutathione peroxidase [60]. The group “ROS and antioxidant processes” (Appendix A) includes 11 genes that were upregulated after exposure to different types of radiation, including six shared by all three types of IR. These encompass an M0XLI1 peroxidase gene and A0A8I6XDH6 glutathione-S-transferase T3-like protein (NAM-associated domain-containing). Plant glutathione-S-transferase proteins (GSTs) have been shown to regulate redox homeostasis by modulating glutathione content and redox state [61]. GSTs also participate in light signalling in *Arabidopsis* [62], in salt tolerance through regulating xylem cell proliferation, ion homeostasis, and reactive oxygen species scavenging in poplars [63], and in plant drought tolerance [64]. The glutathione-S-transferase T3-like protein contains the NAM-domain (No Apical Meristem) and showed discrete localisation within the nucleus in *Arabidopsis*, possibly serving a role in reducing nucleic acid hydroperoxides or in signalling [65]. A NAM domain gene, *GhNAC79*, improves resistance to drought stress in upland cotton [66]. NAM is a part of NAC and CUC2 transcription factors, constituting one of the largest groups of plant-specific transcription factors, widely involved in signalling in response to multiple abiotic stresses [67,68]. Our study revealed the upregulation of another NAC domain-containing protein, F2D6W9 (Table 1), for all IR types.

The upregulated genes shared by the three types of IR also involve two laccase genes (*F2DXF2* and *F2D9J2*, Table 1), and these multicopper oxidase enzymes catalyse a range of oxidative reactions of aromatic and non-aromatic compounds related to cell wall functioning and lignin biodegradation [69,70]. Laccase involvement in plant response to other types of abiotic stress has been reported: *PeuLAC2* in *Populus euphratica* improves drought tolerance by enhancing water transport capacity, and lines with overexpression of *PeuLAC2* exhibit a stronger antioxidant response and greater drought tolerance than wild-type plants [71]; the citrus laccase gene *CSLAC18* contributes to cold tolerance [72]; gene expression analysis of *Aeluropus littoralis* laccases reveals their induction in response to abiotic stresses such as cold, salt, and osmotic stress, as well as ABA treatment [73].

Another upregulated by the three types of IR ROS-related genes is *M0XEM7* (Table 1), encoding a CDI (Cd^2+^-induced) protein (nucleotide-diphospho-sugar transferase), which is required for pollen germination and tube growth, involving cell wall biosynthesis and modification, in *Arabidopsis* [74]. The *Chlamydomonas reinhardtii* gene, encoding a protein with strong similarity to *Arabidopsis* proteins of the nucleotide-diphospho-sugar transferases superfamily, is important for acclimation to phosphorus and sulphur deprivation [75].

A0A287KQB5 encodes for germin-like protein 5-1 (GLP, Table 1). GLPs were upregulated during the early somatic embryogenesis and responded to high temperature stress and 2,4-D treatment [76]. Moreover, overexpression of *Dimocarpus longan* GLP1-5-1 in the globular embryos of longan promoted lignin accumulation and decreased the H_2_O_2_ content by regulating the activities of ROS-related enzymes [76].

Special attention should be given to the gene *A0A8I6WYP3*, which encodes a protein with disulfide reductase activity (thioredoxin domain-containing protein). The gene was significantly upregulated only in samples exposed to protons and electrons (Appendix A). A higher expression level of this gene may indicate increased oxidative stress, as thioredoxin isoforms and NADPH-dependent thioredoxin reductase C (NTRC) act as redox regulatory factors involved in multiple plastid biogenesis and metabolic processes [77] and offer protection against oxidative damage [78].

Additionally, the upregulation of the gene *A0A8I6WG88*, encoding aquaporin, is noted in plants irradiated by electrons and protons. Aquaporins are membrane channels that facilitate water transport and small neutral molecules across biological membranes in most living organisms [79]. They also play a key role in hydraulic regulation in roots and leaves during drought, as well as in response to various stimuli such as flooding, nutrient availability, temperature, or light [80,81,82]. Aquaporins can also facilitate hydrogen peroxide transport [83] and are known to be upregulated under chronic irradiation conditions [84]. The increase in their activity in our data is associated with the upregulation of ROS-related responses (Appendix A).

The direct and indirect (through ROS generation) action of ionising radiation results in DNA damage [85]. We identified 3 upregulated genes (Appendix A) involved in DNA repair, namely *A0A8I6WGY7* (encoding SAP domain-containing protein), which was actively expressed after proton and electron exposure; *A0A8I6YZJ2*, associated with adenosine 5′-diphosphate, was significantly upregulated under all types of exposure; F2E6W2, encoding SWIB domain-containing protein; p53 negative regulator-like.

The cell wall appears to be one of the main targets of ionising radiation exposure [15], reacting to direct ionisation and subsequent ROS exposure. The upregulation of the cellulose synthases *D9IXC7* and *F2DMG1* genes, as well as *A0A8I6X4I5* beta-galactosidase, was noted (Table 1, Appendix A). Recent advancements demonstrate the tight regulation of cellulose synthesis and microtubule arrangement at the primary cell wall by phytohormone networks under stress [86]. The expression of beta-galactosidases is also phytohormone-dependent [87].

It is known that phytohormonal balance changes after radiation exposure [27]. In our data, we specifically identified two upregulated genes involved in auxin biosynthesis and signalling (Appendix A): *A0A8I6YN81* (proton and electron radiation) and *A0A8I6XJC3* (electron and gamma radiation). Research shows that auxins play a critical role in regulating the effects of plant stress [88]. The dynamics and differential distribution of the auxin in plant tissues control an impressive variety of developmental processes that adapt plant growth and morphology to environmental conditions. Various ecological and endogenous signals can be integrated into changes in auxin distribution through their influence on local auxin biosynthesis and intercellular auxin transport [89]. In response to ionising radiation, auxin dynamics is associated with radiation hormesis, a phenomenon of growth stimulation after low-dose radiation exposure [27]. Besides plant development and stress response processes, auxin signalling is involved in DNA repair mechanisms and cell cycle arrest [90].

Photosynthesis is the main energy source for plants, and any abiotic stress affecting this process leads to a cascade of reactions to ensure the proper functioning of the photosynthetic apparatus, and ionising radiation is no exception [6]. We identified two genes, *F2CWQ1* and *A0A8I7BCP4*, as transcription factors involved in photosynthesis, which are actively expressed under all types of radiation. Two other photosynthesis-related genes, *A0A8I6YCH8*, involved in electron transfer and the chaperone gene *F2DDU3* [91], were upregulated only after proton and electron exposure.

Theoretical crosstalk of the products of IR-responsive transcripts, revealed in this work, and studies of other abiotic stressors are schematically presented in Figure 3.

## 4. Materials and Methods

An extended multiomics experiment involving ionising radiation with low- and high-LET exposure of winter barley (*Hordeum vulgare* L.) seedlings was performed [40]. The specific subset of transcriptomic data related to low-LET IRs was analysed in detail and presented in the current article.

### 4.1. Growing Conditions and Sampling

Grains of the cultivar Fox 1 were planted in pots with 120 g of soil (macronutrients (mg × L^−1^): N—100, P_2_O_5_—50, K_2_O—200, MgO—30; trace elements (mg × kg^−1^): Cu—3.0–5.0; Zn—0.2–0.3; Mn—8.0–40.0; Fe—0.8–2.0; Mo—0.1–0.4; B—0.4–1.6; pH—5.5–6.5, according to the manufacturer) at a depth of 1.5–2 cm and watered with 50 mL of H_2_O. Five seeds were sown into each pot. The growth chamber “Fitotron LiA-2” (OOO “Fitotron”, Zelenograd, Russia) was used for plants germination and development under 21 °C and 50% humidity, continuous LED light (set of wavelengths 440, 460, 525, 620, and 660 nm), 150–300 µmol photons m^−2^ s^–1^ (depending on plant height).

The seedlings were irradiated on the 7th day after sowing at the stage of the first true leaf unfolded, having a height of 5–7 cm. Control plants underwent the same growing and transportation conditions as irradiated plants. Twenty-four hours after irradiation, the leaves of both control and irradiated plants were harvested. Plants from the same pot were pooled into one sample. Twelve pooled samples (3 biological replicates per condition: control, γ-, electron, and proton irradiations) were used for transcriptome analysis.

### 4.2. Irradiation Conditions and Dosimetry

For irradiation, the facilities of A. Tsyb Medical Radiological Research Centre (Obninsk, Russia) were used. Preliminary tests showed that an absorbed dose of 15 Gy was tolerable for barley juvenile plants and did not lead to growth arrest [92].

The γ-radiation source “Agat” (^60^Co isotope, Scientific Research Institute of Technical Physics and Automation, Moscow, Russia) was used for γ-irradiation (γ) at a dose of 15 Gy. The pots were placed in the 12 × 12 cm^2^ radiation field to cover the leaves and the root system. A NOVAC11 linear electron accelerator (New Radiant Technology, Milan, Italy) was applied for irradiating seedlings with 8 MeV electrons (e^−^) at a dose of 15 Gy. Pots were placed in a beam of electrons with a diameter of 10 cm to expose both leaves and root system. Proton irradiation (p^+^) was performed at the Prometheus Proton Therapy Complex (JSC-PROTOM, Oryol, Russia), capable of accelerating protons in the 30–250 MeV energy range with an average output beam current of 5 × 10^8^ protons per cycle. The pots were placed in the isocentre of the setup opposite each other. The irradiation was performed on the initial phase of the Bragg curve at 100 MeV energy at a dose of 15 Gy, fluence 1.2 × 10^10^ proton/cm^2^.

Dosimetry was performed using a PTW Farmer chamber type 30013 (PTW-Freiburg GmbH, Freiburg im Breisgau, Germany) and a PTW Unidose webline electrometer in a water-equivalent solid-state phantom PTW RW3 Slab (PTW-Freiburg GmbH, Germany). Measurements were executed according to TRS 398 recommendations [42]. The heterogeneity of the irradiation field was checked using 2D array of PTW Octavius 1500XDR ionisation cameras (PTW-Freiburg GmbH, Germany) and did not exceed 5% in the irradiation area in all directions (3% for proton irradiation).

### 4.3. Transcriptome Analysis

#### 4.3.1. RNA Isolation, Library Preparation, Illumina Sequencing

Total RNA was isolated using the GeneJET RNA Purification Kit (Thermo Fisher Scientific, Cleveland, OH, USA) with polyvinylpyrrolidone addition (Sigma-Aldrich, Darmstadt, Germany). The quality and purity of isolated RNA were assessed using the NanoDrop OneC (Thermo Fisher Scientific, USA) and horizontal gel electrophoresis. cDNA synthesis, library preparation, and sequencing were provided by the “Evrogen” (Moscow, Russia). Using the TruSeq mRNA Stranded reagent kit (Illumina, San Diego, CA, USA), poly(A+)-fraction enrichment and random primer cDNA synthesis were performed in 12 RNA samples (four conditions (γ-irradiated, p^+^-irradiated, e^−^-irradiated and non-irradiated as control) × three replicates). The cDNA was used as templates for Illumina sequencing technology libraries. The libraries’ quality was tested using Fragment Analyzer (Agilent, Santa Clara, CA, USA). Quantitative analysis was accomplished using the qPCR method. The cDNA libraries were sequenced at Illumina NovaSeq 6000 (2 × 150 bp). FASTQ files were acquired using bcl2fastq v2.20 Conversion Software (Illumina, USA). As a result, 1,183,039,344 raw reads were received: 299,026,364 reads for e^−^-irradiated samples, 310,498,678 reads for γ-irradiated samples, 297,436,970 reads for p^+^-irradiated samples, and 276,077,332 reads for non-irradiated (control) samples.

#### 4.3.2. Data Processing

Data pre-processing, including quality control, screening of sequence library against a set of reference sequence databases, and filtering, was performed on Debian 10.13 releases. The quality control checks of raw sequence data were conducted using FastQC v 0.11.9 and MultiQC v 1.11 [93]. The search for the received sequence dataset against a panel of different genomes to determine the sequences’ origin was performed using Fastq Screen v 0.14.0. Afterwards, low-quality reads were filtered with Trimmomatic v 0.39 [94] for paired-end data, using ILLUMINACLIP, HEADCROP, LEADING, TRAILING, and MINLEN parameters.

The filtered reads were quasi-mapped to barley reference transcriptome (Barley Reference Transcript Dataset (BaRTv2.18) [95]) using Kallisto v 0.46.1 [96]. The results of differential gene expression for the obtained TPM (transcripts per million reads) values were obtained with DESeq2 v 1.38.3 [97] using the negative binomial distribution and were shown as the logarithm of fold change (log_2_FC). The accepted level of significance was 0.05 (attained by the Wald test, corrected for multiple testing using the Benjamini–Hochberg method). Only genes showing log_2_FC > |1| were considered as differentially expressed.

faSomeRecords v 1.0 was used to obtain FASTA records for all differentially expressed genes (DEGs) from BARTv2.18. These sequences were subsequently utilised for BLASTx analysis with the UniProt database to gain additional gene information using the pseudomolecule assembly MorexV3 (release 54). After analysing DEGs common to the three types of radiation applied, the unidentified proteins were examined, and their closest homologues were identified manually using the NCBI database.

Functional enrichment analysis was performed using the Gene Ontology (GO) database with ShinyGo v 0.75c for the MorexV3, and the visualisation was performed using the agriGO v 2. The accepted significance level was 0.05 (applying Fisher as a statistical test method).

## 5. Conclusions

By examining the gene expression patterns after different types of radiation exposure, we observed similarities in the transcriptional responses to other abiotic stressors. Considering the diverse mode of action of different IRs, we have conducted pairwise comparisons of the expression profiles after exposure and have identified differences that can primarily be attributed to the nature of the ionising radiation used. The genes commonly regulated after exposure to three different types of IR contribute to our understanding of the overall plant response to ionising radiation. Considering the similarity in transcriptional responses to different stressors, we identified which DEGs were known to be involved in responses to other stressors, providing insights into the universal mechanisms of response to abiotic stressors. The obtained data allow us to hypothesise that the majority of genes with high expression levels after exposure to three types of ionising radiation not only help plants survive radiation stress but also other abiotic stressors, especially with genotoxic components (such as heavy metals and drought). They include a set of ROS-responsive genes (Uniprot IDs of proteins: M0XLI1, A0A8I6XDH6, F2D6W9, F2DXF2, F2D9J2, A0A287KQB5, A0A8I6WYP3, and A0A8I6WG88); genes involved in DNA repair (A0A8I6WGY7, A0A8I6YZJ2, and F2E6W2); cell wall metabolism-related genes (D9IXC7, F2DMG1, and A0A8I6X4I5); genes involved in auxin biosynthesis and signalling (A0A8I6YN81 and A0A8I6XJC3), and photosynthesis-related genes (F2CWQ1, A0A8I7BCP4, A0A8I6YCH8, and F2DDU3). These molecules can be plausible targets for creating barley cultivars resilient to multiple stress exposures.

## Figures and Tables

**Figure 1 plants-13-00342-f001:**
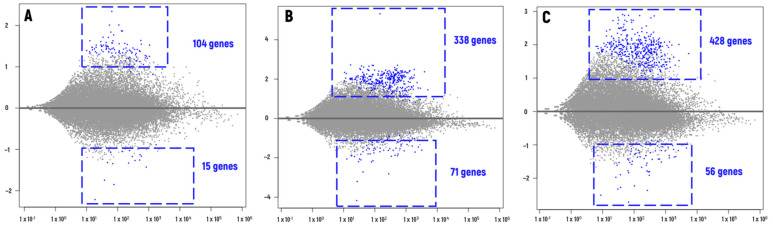
Overview of differential gene expression for gamma (**A**), proton (**B**), and electron (**C**) irradiated seedlings of *H. vulgare* in comparison with the non-irradiated samples. The plots were created using the DeSeq2 Packages for RStudio v 1.4 (R-Tools Technology, Richmond Hill, ON, Canada). Graphical changes were made with Microsoft PowerPoint 2019 (Microsoft Corporation, Redmond, WA, USA). Y-axis—log_2_FC values; X-axis—mean of normalised counts.

**Figure 2 plants-13-00342-f002:**
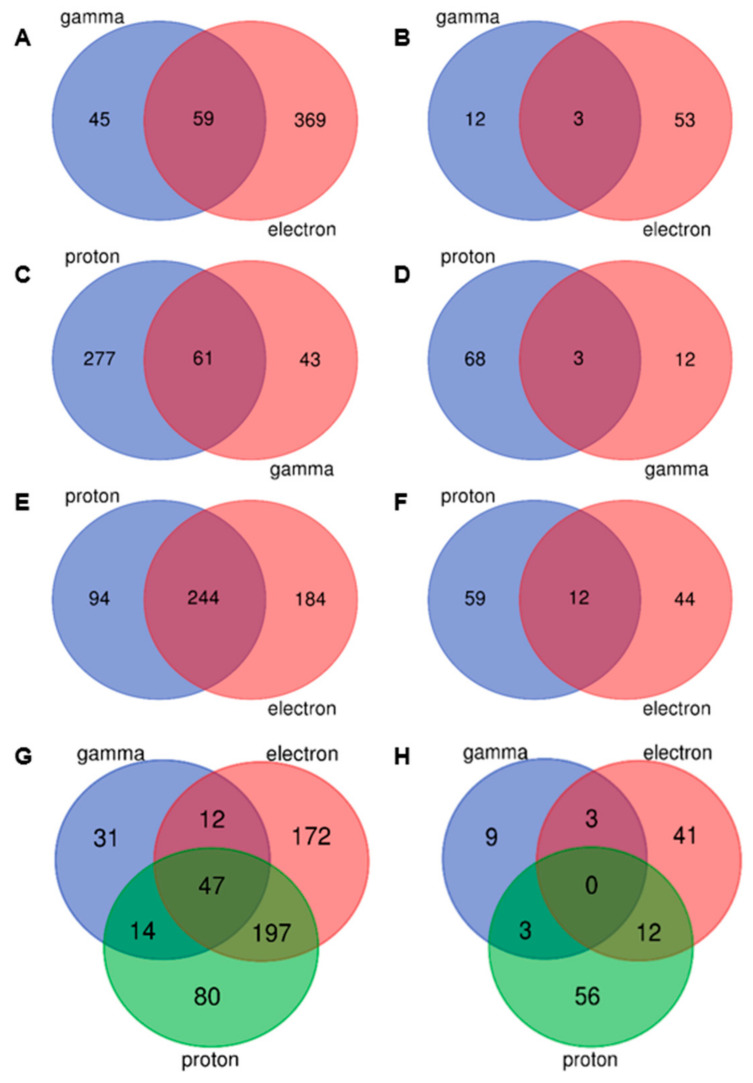
Venn diagrams reflecting unique and common up- (**A**,**C**,**E**,**G**) and downregulated (**B**,**D**,**F**,**H**) genes 24 h after gamma and electron (**A**,**B**); proton and gamma (**C**,**D**); proton and electron (**E**,**F**) exposures; all 3 types of irradiation (**G**,**H**).

**Figure 3 plants-13-00342-f003:**
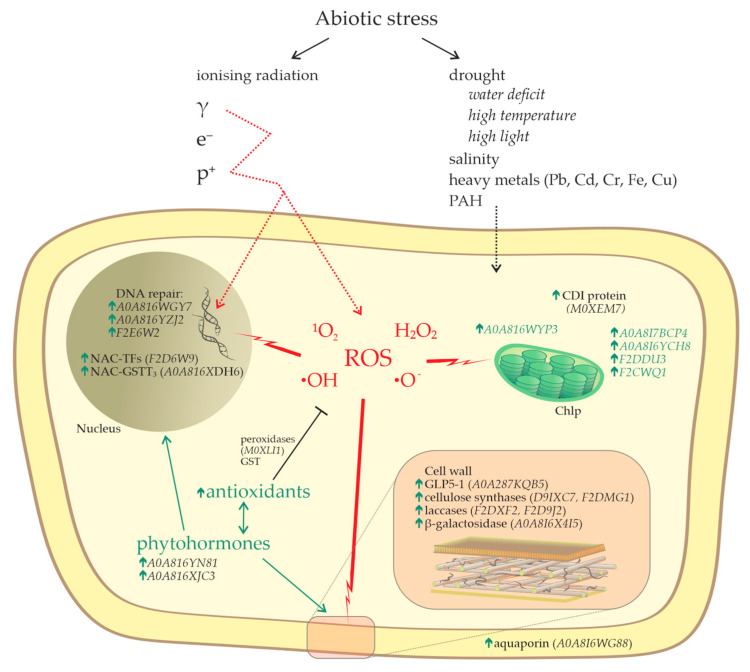
Schematic representation of the revealed molecular players in plant cell responses to IR and other abiotic stress factors.

**Table 1 plants-13-00342-t001:** DEGs with the common expression pattern after irradiation with γ-rays, electrons, and protons.

Gene	Uniprot ID	Protein	log_2_FC		
			γ	e^−^	p^+^
CELL CYCLE					
*BaRT2v18chr2HG061030* *HORVU.MOREX.r3.2HG0116520*	A0A8I7B533	Chloroplastic pseudouridine synthase 2	1.52	2.67	2.69
SIGNAL TRANSDUCTION					
*BaRT2v18chr1HG005570* *HORVU.MOREX.r3.1HG0012160*	A0A287EMU9	Cysteine-rich protein kinase	1.38	2.86	1.71
*BaRT2v18chr1HG026150* *HORVU.MOREX.r3.1HG0057590*	F2CXV3	Nucleoside diphosphate kinase	1.64	2.06	2.39
METABOLISM					
*BaRT2v18chr1HG008880*	Q96565	3-aminomethylindole N-methyltransferase	1.06	1.92	2.03
*BaRT2v18chr4HG198110*	-	ATP-dependent Clp protease proteolytic subunit	1.18	1.78	1.83
*BaRT2v18chr5HG235800* *HORVU.MOREX.r3.5HG0456510*	A0A8I7B8L0	Dihydrolipoamide acetyltransferase, component of pyruvate dehydrogenase complex	1.14	1.67	1.79
*BaRT2v18chr3HG132390* *HORVU.MOREX.r3.3HG0252270*	A0A287KQB5	Germin-like protein 5-1; Cupin type-1 domain-containing protein	1.87	2.27	2.19
*BaRT2v18chr3HG141240* *HORVU.MOREX.r3.3HG0274060*	A0A287L592	Aspartic proteinase oryzasin-1	1.37	2.74	2.44
*BaRT2v18chr6HG320720* *HORVU.MOREX.r3.6HG0620630*	A0A8I6Y7Y0	Subtilisin-like protease	1.37	2.09	2.53
CELL WALL					
*BaRT2v18chr1HG017980* *HORVU.MOREX.r3.1HG0041280*	D9IXC7	Cellulose synthase	1.80	1.70	1.70
*BaRT2v18chr1HG036030* *HORVU.MOREX.r3.1HG0073500*	F2D9J2	Laccase	2.34	2.06	2.24
*BaRT2v18chr3HG149100* *HORVU.MOREX.r3.3HG0288960*	F2DMG1	Cellulose synthase	1.15	1.47	1.38
*BaRT2v18chr3HG157600* *HORVU.MOREX.r3.3HG0302570*	F2DXF2	Laccase	1.33	2.14	1.84
*BaRT2v18chr4HG210440* *HORVU.MOREX.r3.4HG0402760*	A0A8I6X4I5	Beta-galactosidase	1.41	2.17	2.19
*BaRT2v18chr6HG284110* *HORVU.MOREX.r3.6HG0540930*	A0A8I7BAT6	Isoflavone reductase	1.53	2.45	1.97
TRANSPORT					
*BaRT2v18chr2HG058330* *HORVU.MOREX.r3.2HG0111570*	A0A8I6WX86	Protein transport protein sec16	1.52	2.29	1.97
*BaRT2v18chr5HG257150* *HORVU.MOREX.r3.5HG0496010*	A0A8I6XRW4	Trigger_N domain-containing protein	1.52	2.07	2.34
PHOTOSYNTHESIS					
*BaRT2v18chr4HG203760* *HORVU.MOREX.r3.4HG0392110*	A0A8I7BCP4	High molecular mass early light-inducible protein HV58, chloroplastic	1.55	1.95	2.10
*BaRT2v18chr7HG352120* *HORVU.MOREX.r3.7HG0680970*	F2CWQ1	Transcription termination factor MTERF6, chloroplastic/mitochondrial	1.33	1.96	1.79
ROS DEFENCE					
*BaRT2v18chr5HG274520* *HORVU.MOREX.r3.5HG0525430*	A0A8I6YN10	PMR5N domain-containing protein	1.45	2.19	2.46
*BaRT2v18chr7HG341270 HORVU.MOREX.r3.7HG0659410*	M0XEM7	Protein CDI	1.44	2.02	2.41
*BaRT2v18chr7HG379410* *HORVU.MOREX.r3.7HG0737470*	M0XLI1	Peroxidase	1.59	2.14	1.93
*BaRT2v18chr6HG304290* *HORVU.MOREX.r3.2HG0120250*	A0A8I6XDH6	Glutathione-S-transferase T3-like protein	1.49	1.74	1.81
CONTROL OF GENE EXPRESSION					
*BaRT2v18chr5HG271560* *HORVU.MOREX.r3.5HG0520550*	A0A8I6YMI0	Protein Argonaute 1-like	1.58	2.31	2.19
*BaRT2v18chr3HG143530* *HORVU.MOREX.r3.3HG0279110*	F2D6W9	NAC domain-containing protein	1.50	1.80	2.14
*BaRT2v18chr5HG266730* *HORVU.MOREX.r3.5HG0512410*	F2E6W2	SWIB domain-containing protein; p53 negative regulator-like	1.33	2.70	2.44
RIBOSOME					
*BaRT2v18chr3HG158290*	-	40S ribosomal protein S23-like	1.34	2.23	2.38
*BaRT2v18chr7HG368730*	-	40S ribosomal protein S24-1	1.39	2.05	2.35
*BaRT2v18chr7HG351350*	-	60S ribosomal protein L31	1.57	2.36	2.43
*BaRT2v18chr5HG268570*	-	60S ribosomal protein L35a-3	1.54	1.72	2.14
*BaRT2v18chr5HG268620*	-	60S ribosomal protein L44	1.45	1.98	2.26
*BaRT2v18chr5HG266050*	-	60S ribosomal protein L13a-4-like	1.35	2.01	2.36
*BaRT2v18chr4HG174880*	-	40S ribosomal protein S25	1.50	1.70	2.25
*BaRT2v18chr3HG143320*	-	60S ribosomal protein L18a-like	1.32	1.91	2.08
*BaRT2v18chr2HG080740*	-	40S ribosomal protein S8	1.25	1.99	2.07
*BaRT2v18chr2HG054760*	-	40S ribosomal protein S6	1.33	2.10	2.35
*BaRT2v18chr1HG014270*	-	40S ribosomal protein S7	1.34	1.95	2.06
*BaRT2v18chr1HG033630*	-	60S ribosomal protein L30	1.38	2.09	2.21
*BaRT2v18chr3HG158010*	-	Putative ribosomal protein S8	1.37	1.94	2.15
*BaRT2v18chr4HG204910* *HORVU.MOREX.r3.4HG0332030*	A0A8I6XW53	Ubiquitin-60S ribosomal protein L40-1	1.46	2.14	2.26
*BaRT2v18chr7HG361220*		60S acidic ribosomal protein P0	1.22	1.72	2.01
UNKNOWN/UNCERTAIN					
*BaRT2v18chr2HG052720* *HORVU.MOREX.r3.2HG0100380*	A0A8I6WJY6	Uncharacterised protein	1.23	1.45	1.52
*BaRT2v18chr2HG108330* *HORVU.MOREX.r3.2HG0206630*	A0A8I6WWY3	Rx_N domain-containing protein	2.01	2.45	2.70
*BaRT2v18chr4HG195150*	-	Hypothetical protein ZWY2020_048147	1.24	1.73	1.95
*BaRT2v18chr6HG327620* *HORVU.MOREX.r3.6HG0632840*	A0A8I6YZJ2	NB-ARC domain-containing protein	1.32	2.06	1.90
*BaRT2v18chr2HG062780* *HORVU.MOREX.r3.2HG0119860*	F2DB07	Chaperonin 10-like	1.23	1.76	1.66
*BaRT2v18chr6HG321270*	-	Hypothetical protein ZWY2020_023007	1.45	2.20	2.36

## Data Availability

The processed RNA-sequencing files are available at the Sequence Read Archive (BioProject PRJNA980246).

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
