# Peer review of "Comparative Analysis of the Effect of Gamma-, Electron, and Proton Irradiation on Transcriptomic Profile of Hordeum vulgare L. Seedlings: In Search for Molecular Contributors to Abiotic Stress Resilience"

_plants, 2024, doi:10.3390/plants13030342_

Round 1

Reviewer 1 Report

Comments and Suggestions for Authors

In this sudy, the authors studied the transcriptomic profiles of barley seedlings after exposure to three different ionising radiation (IR, γ-rays, electrons, and protons). Among IRs, the highest number of differentially expressed genes was observed in electron-irradiated samples (428 upregulated and 56 downregulated genes). It was reported that expose resulted in upregulation of ROS-responsive genes, genes involved in DNA repair, cell wall metabolism, auxin biosynthesis and signalling, as well as photosynthesis-related genes.

Author Response

Dear Reviewer,

Thank you very much for taking the time to review this paper and for the high estimate of the manuscript!

with the very best wishes,

the authors of the manuscript

Reviewer 2 Report

Comments and Suggestions for Authors

The manuscript presented comparative analysis of  the transcriptomic profiles of barley seedlings after exposure to γ-rays, electrons, and protons. The design, treatments, sampling and data analysis were all carried out properly.  The writing is well.

One minor thing needs to be clarified - seedlings obtained after sowing 7 days, how many leaves do they have? the height?

Author Response

Dear Reviewer,

Thank you very much for taking the time to review this paper and for the high estimate of the manuscript!

Please find the detailed response below and the corresponding corrections highlighted in the re-submitted files.

best regards,

the authors of the manuscript

Reviewer 3 Report

Comments and Suggestions for Authors

This paper investigates the effects of three types of irradiation on the transcriptome response in barley. The authors conducted a detailed study of the differentially expressed genes after treatment. This is a topic of interest to researchers in related fields, but the paper needs some improvement before acceptance for publication. My detailed comments are as follows:

1. According to the Discussion section, the authors suggest that the molecular pathway of IR response may be related to the molecular pathway of abiotic stress response in plants, as genes significantly up-regulated after irradiation are involved in plant ROS response to other abiotic stresses. Is it possible to provide more concrete evidence, e.g. by carrying out experiments, etc.?

2. Add header information to the supplementary tables.

3. 2.1 and above in Result can be merged into one section and retitled.

4. In Result section 2.3 “The total number of commonly expressed genes under proton and electron irradiation was 256”. How is 256 calculated.

Comments on the Quality of English Language

Good

Author Response

Dear Reviewer,

Thank you very much for taking the time to review this paper, for thoroughly checking the manuscript and for the valuable comments and suggestions.

Please find the detailed response below and the corresponding corrections highlighted in the re-submitted files.

best regards,

the authors of the manuscript
